

# 1 Carbon, nitrogen and sulfur (CNS) status and dynamics in
# 2 Amazon basin upland soils, Brazil

Jörg Matschullat[1], Roberval Monteiro Bezerra de Lima[2], Sophie F. von Fromm[1,3], Solveig Po-
spiech[4], Andrea M. Ramos[5], Gilvan Coimbra Martins[2], Katharina Lenhart[6]
[1]Interdisciplinary Environmental Research Centre, TU Bergakademie Freiberg, Brennhausgasse 14, 09599
Freiberg, Germany; email: matschul@tu-freiberg.de; ORCID: 0000-0003-0549-7354
[2]Embrapa Amazônia Ocidental, Rodovia AM 10, km 29, s/n, Manaus, AM, CEP 69010-970, Brazil; email: rob-
erval.lima@embrapa.br; ORCID 0000-0003-1260-447X
[3]Max-Planck-Institute for Biogeochemistry, P.O. Box 100164, 07701 Jena, Germany; email: sfromm@bgc-
jena.mpg.de; ORCID 0000-0002-1820-1455
[4]Helmholtz Institute for Resource Technology, Chemnitzer Str. 40, 09599 Freiberg, Germany; email:
s.pospiech@hzdr.de; ORCID 0000-0003-2727-2375
[5]National Institute for Meteorology INMET, St. Sudoeste-Brasília, DF, CEP 70680-900, Brazil; email: an-
drea.ramos@inmet.gov.br; ORCID 0000-0001-7414-3395
[6]Technische Hochschule Bingen, Berlinstraße 109, 55411 Bingen am Rhein, Germany; email: k.lenhart@th-
bingen.de; ORCID 0000-0001-5226-492X
Correspondence to: Jörg Matschullat (matschul@tu-freiberg.de)
**Key Points:**
• Quantification of Amazon basin upland soil CNS concentrations and $C_{org}$/N ratios
• CNS median concentrations are similar to European soils; ranges are more narrow
• In up to 50 years after deforestation, average $C_t$, $C_{org}$ and N losses of up to 20 % in post-forest
soils
• Intra-annual dynamics of soil pH-values and electrical conductivities under generally acidic to
strongly acidic conditions and very low λ
• Central Amazonas soils show stronger differences between forest and post-forest land than
southern Amazonas soils – indicator of significant impact and edge effects in the south?
**Key words**: tropical soils; terra firme; oxisol; ferralsol; soil biogeochemistry; climate change; nutrient ratios,
hydrological gradient
**Abstract.** Given the dimensions of the Amazon basin (7.5 million km[2]), its internal dynamics, increasing anthro-
pogenic strain on this large biome, and its global role as one of two continental biospheric tipping elements, it
appears crucial to have data-based knowledge on carbon and nitrogen concentrations and pools as well as on
possible intra-annual dynamics. We quantified carbon ($C_t$, $C_{org}$), nitrogen (N) and sulfur (S) concentrations in litter
(ORG) and mineral soil material (TOP 0–20 cm, BOT 30–50 cm) of upland (terra firme) oxisols across Amazo-
nas state and present a first pool calculation. Data are based on triplicate seasonal sampling at 29 sites (forest and
post-forest) within the binational project EcoRespira-Amazon (ERA). Repeated sampling increased data accura-
cy and allows for interpreting intra-annual (seasonal) and climate-change related dynamics. Extreme conditions
between the dry season in 2016 and the subsequent wet season (ENSO-related) show differences more clearly.
Median CNS in the Amazon basin TOP soils ($C_t$ 1.9, $C_{org}$ 1.6, N 0.15, S 0.03 wt-% under forest canopy) as well as
$C_{org}$/N ratios show concentrations similar to European soils (FOREGS, GEMAS). TOP $C_t$ concentrations ranged
from 1.02 to 3.29 wt-% (median$_{Forest}$ 2.17 wt-%; median$_{Post-Forest}$ 1.75 wt-%), N from 0.088 to 0.233 wt-% (median$_{Forest}$
0.17 wt-%; median$_{Post-Forest}$ 0.09 wt-%) and S from 0.012 to 0.051 wt.-% (median$_{Forest}$ 0.03 wt.-%; median$_{Post-Forest}$ 0.02 wt-





%). C$_{org}$/N ratios ranged from 6 to 14 (median 10). A first pool calculation (hectare-based) illustrates forest versus post-forest changes. The elements are unevenly distributed in the basin with generally higher CNS values in the central part (Amazonas graben) as compared to the southern part of the basin. Deforestation and drought conditions lead to C and N losses – within 50 years after deforestation, C and N losses average 10 to 15 %. Regional climate change with increased drought will likely speed up carbon and nitrogen losses.

## 1 Introduction

The Amazon basin is seen as one of the terrestrial tipping elements (Lenton et al., 2008) with global impacts to be expected when triggered (Reid et al., 2015). Several works suggest that the Amazon rainforest biome may tip by the end of this century, and give rise to a subsequent, more dry forest environment (Shukla et al., 2008). Recent changes in the basin (Davidson et al., 2012; Fearnside, 2018; Hubbell et al., 2008; Longobardi et al., 2016; McGrath et al., 2001; Rosa et al., 2016; Shukla et al., 2008), including increasing extreme events such as drought (e.g., dry season 2016) with strong impacts on the hydrological cycle far beyond the basin (Aragão, 2012; Davidson et al., 2012; Jiménez-Muñoz et al., 2016; Lewis et al., 2011; Marengo, 2006) corroborate that hypothesis. It is in this light that evidence-based knowledge and reliable data are needed to better assess possible resilience and consequences of current land use in the basin. Nagy et al. (2016) compiled related state-of-the-art of knowledge in biosphere-atmosphere-human land use interactions within the Amazon basin, updating and expanding previous work (e.g., McClain et al., 2001). Cerri et al. (2006a) pointed out the knowns and unknowns of land conversion in Latin America with particular emphasis on soil carbon sequestration, with Cerri et al. (2006b) breaking this attempt down to the Amazonian tropical rainforest biome. One conclusion is that reliable data are still scarce. Nitrogen data are particularly missing. Given the relevance of the carbon and nitrogen cycles in global change, and the diverse estimates of related soil pools (e.g. Baldock, 2007; McNeill and Unkovich, 2007) it appears particularly valuable to obtain better data. Various questions remain unanswered to better understand carbon, nitrogen and sulfur (CNS) cycling in humid tropical forest regions, and particularly about consequences of human impact in such systems. Key questions for this work can thus be formulated:

What are CNS concentrations in Amazon basin soils – and how do these compare to other world soils? Are there differences in CNS status between forest and post-forest land? Is the CNS status similar throughout the year or do we see intra-annual dynamics? How do mineral soil CNS concentrations relate to the organic litter layer? Is the hydrological difference between the more humid central part of the basin and its southern "shoulder" relevant? Which consequences may future increasing drought conditions have for CNS behaviour in Amazon basin soils?

Answers to these questions are used for a first upscaling attempt that also looks into the near future. Within the binational project EcoRespira-Amazon (ERA: https://blogs.hrz.tu-freiberg.de/ecorespira/), we studied the geochemistry of upland (terra firme) oxisols in Amazonas state, Brazil. It shows the CNS status and its intra-annual dynamics. The latter is based on our observations under extreme dry season and under wet season conditions during El Niño-Southern Oscillation (ENSO) and post-ENSO dominance (Jiménez-Muñoz et al., 2016).

Soil diversity in the tropics is high (Sanchez, 2019: 71f). Humid tropical soils such as those dominating the Amazon basin are often reported to be nutrient-depleted (Richter and Babbar, 1991), even referred to as 'infertile soils' (Irion, 1978). This generalization may be partly true for base cations that are easily leached under the dominant high to very high soil acidity and plenty of water (Juo and Franzluebbers, 2003). Hydrolysis of primary



minerals there is about 100 times higher as compared to temperate climate soils (Glaser et al., 2004). The oxisols
(ferralsols) in our study fall into this category (Juo and Franzluebbers, 2003). Other components including CNS
may not follow such simplified vision. Sanchez (1977) already published average organic carbon values of 2.01
wt.-%. Bohn et al. (2001: 156f) interpret that as a result of strong interactions between soil organic matter
(SOM) with Fe and Al hydroxy-oxides and allophane, which could stabilize SOM against microbial decay. At
the same time, they acknowledge "*the high rate of year-round biomass production in the humid tropics*", which
counteracts the relative nutrient paucity in these soils. Previous works usually deliver spatially-clustered data;
sometimes focussing on very large-scale soil variations (Quesada et al., 2010). A closer look at intra-basin phys-
ico-chemical variability appears necessary and helpful in order to better understand CNS dynamics and soil
fertility in this potentially vulnerable system. Pre-Columbian tropical agricultural practices, such as "Indigenous
Black Earth" (terra preta do índio) demonstrate that significant carbon and nitrogen enrichment is possible under
these climatic conditions and that long-term steady state can be reached through appropriate land-use techniques
(Glaser and Birk, 2012). These historical anthroposols may exemplify sustainable agriculture, to be frequently
encountered near old settlements (Roosevelt, 2013; Sanchez, 2019: 285f; own observations).
Litter (ORG) and mineral soil (TOP and BOT) was collected to assess their element concentrations at the study
sites. Repeated sampling of the same sites under different seasonal conditions served both quality control and to
investigate seasonal dynamics, based on our observation of radically changing litter layer thickness between dry
and wet season. Following a presentation of pH-values, electrical conductivity and soil colour, the CNS results
in ORG, TOP and BOT material are discussed in detail. An attempt is made to upscale results to upland Amazon
soil for the entire basin.

## 2       Materials and methods

### 2.1       Site characteristics

We sampled at 29 sites from 13 upland (terra firme) locations within Amazonas state, Brazil, covering an area of
roughly 930 by 800 km (740,000 km²). Each location (Figure 1) represents 2–3 sites with different land cover; at
least one forested (F), the other pastureland, plantation or agroforestry (post-forest sites, PF). The native forest
locations showed some (mostly minor) human impact; their conjunct areas were considerably larger in the cen-
tral part as compared to the south. To represent much of Amazon basin upland soils and their land cover, a corre-
sponding number of sites was studied in the central part of the basin and across southern Amazonas state (Figure
1). Soils are generally fully oxidized on the macro scale (Sanchez, 1977). They were classified as oxisols (or
ferralsols or latosols depending on classification type; Batjes, 2016; Ross, 2006; Sanchez, 2019). Hydrolysis
under oxidizing acidic conditions is the dominant weathering agent (Juo and Franzluebbers, 2003).
Average location altitudes range from 25 m a.s.l. (06 Manacapuru) to 190 m a.s.l. (07 Apuí and 12 Nova
Califórnia) with a median altitude of 90 m a.s.l. (mean: 105 m a.s.l.). The central part of the Amazon basin (So-
limões-Amazon Graben) is characterized by lower average altitudes (median 75 m a.s.l., mean 80 m a.s.l.) and
younger lithological material; mostly Cretaceous, Tertiary and Quaternary sediments (CPRM 2005). Deforesta-
tion is still moderate, albeit rather substantial near the selected locations. Six of the 13 locations are located here
(Figure 1: 01–06), all part of the Eastern Amazon Basin plateau (Ross, 2006). The other seven locations (07–13)
represent the southern "shoulder" of the basin, the 'Marginal depression of southern Amazônica' (Ross, 2006)



with a median elevation of 150 m a.s.l. (mean 130 m a.s.l.). Geologically, this is part of the older Amazonian
Craton with partly surfacing Paleoproterozoic, Silurian and Devonian rocks and some Tertiary and Quaternary
sediments (CPRM 2005). Deforestation here is partly radical (> 90 %).
Average annual temperature in the study area is 27 °C with maxima in September and October (28 °C) and min-
ima from January to March (26.3 °C). The mean annual number of days with maximum monthly and annual
temperatures exceeding 30 °C is 300 with the highest average in August and October (29 days) and the lowest in
February (20 days). Sunshine hours average 1828.5 annually with monthly values exceeding 200 hours in July,
August and September. Average atmospheric pressure in Manaus is 1003.7 millibar (PMSL 1010.7 mbar) with
minima in November (1002.6 mbar; PMSL 1009.2 mbar), rising gradually until July (1005.4 mbar; PMSL
1012.5 mbar). Average annual precipitation amounts to 2301.2 mm with the driest month being August (64.3
mm). April is the wettest month with 319.0 mm. The mean annual number of days with precipitation above or
equal to 1 mm is 155, with the highest average in January and March (19 days) and the lowest in September (6
days). The mean annual number of days with precipitation above or equal to 10 mm is 73 days, with the highest
average in April (10 days) and the lowest in August (2 days). With 15 mm threshold, the mean annual number of
days is 54 with the highest in April (8 days) and the lowest in August (1 day). Relatively high air humidity oc-
curs all year round (avg. 82.8 %) with minima in September (72.2 %) and maxima in March (86.9 %). Annual
mean prevailing wind direction is Southeast, followed by north-easterly air streams. Average annual potential
evapotranspiration is 192 mm with maxima in October (234.4 mm) and minima in February (167.1 mm).
These average data (climate normal of 1981–2010) see some shift, when comparing them with data from the past
(CLINO 1961-1990; electronic supplement Table A01). Conditions during the sampling time (2016 and 2017)
were about 38 % dryer in the central part and 66 % more dry in the southern part, compared with the climate
normal 1981–2010. The southern region is generally dryer then the central one. All information is based on sta-
tion data from the INMET network (electronic supplement Table A01).
**2.2    Soil sampling**
Ninety percent of the ERA sites (minimum one hectare each; n = 26) were sampled in triplicate (February and
March 2016, weak wet; July and August 2016, extremely dry; February and March 2017, normal wet) to obtain a
robust database. Locations 04 and 10 were sampled a fourth time in February 2018 to compensate previous ac-
cess problems. All sampling spots were geo-referenced (handheld GPS/GLONASS receivers Garmin GPS 64s)
with an accuracy of ± 3 m even under dense forest canopy. Litter material (ORG) was collected across the entire
site with latex glove-protected hands into thoroughly cleaned cotton bags and closed tight until further pro-
cessing. Mineral soil was sampled manually by soil auger (Sondaterra, Piracicaba, São Paulo State, Brazil) in
two different depths (TOP: 0–20 cm, n = 79; BOT: 30–50 cm, n = 80) into Rilsan® polymer sampling bags after
carefully removing surface litter, visible root material and small stones. Three spots were sampled at each site.
Resulting samples (3 x TOP, 3 x BOT) were united to composite TOP and BOT samples per site. The auger was
thoroughly cleaned between all individual samples. This procedure was repeated in all three campaigns. Use of
disposable latex lab-gloves further reduced contamination risks during sampling. Soil humidity was measured in
the field (TDR sensor, Delta-T Devices, United Kingdom).
**2.3    Sample processing, soil solution pH, electrical conductivity and soil colour**



Litter and mineral soil material were air-dried in the lab, crushed in an agate mortar and pistil, and (nylon) sieved
to particles < 2 mm. Larger material (stones, root material) was discarded. The fine material was ground and
milled to analytical size (< 63 $\mu$m) in a planetary ball mill (Pulverisette 7, Fritsch, Idar-Oberstein, Germany) and
a rotating disc mill (RS 200, Retsch, Haan, Germany); both with agate grinding jars and balls. Sample powder
was tested for effective grain size by sieving through nylon test sieves. All equipment was thoroughly cleaned
after each sample.
Soil solution electrical conductivity ($\lambda$ in $\mu$S cm$^{-1}$) and pH-values (pH$_{H2O}$ and pH$_{CaCl2}$) were determined in the soil
fraction < 2 mm. For each sample, 2-times 15.0 g of soil were weighed into two beakers (double determination)
to be filled with 75 mL of deionized water and stirred for 1 hr. Conductivities were determined with calibrated
'cond 3110' electrodes (Meinsberg, Germany). In the same solution and a 0.01 mol CaCl$_2$ solution, pH was
measured. Following 1 hr of sedimentation time, pH-values were determined in the supernatant solution with
calibrated EGA 161 electrodes (Meinsberg, Germany), adapted to low-conductivity solutions.
Soil colours were determined from air-dried samples (< 2 mm) under constant light conditions using the Munsell
soil colour charts (1992) with reference to hue, value and chroma (electronic supplement Table A02).
**2.4    Elemental analysis (CNS)**
Twenty (20.0) milligrams of analytical grade ORG, TOP and BOT aliquots of each pulverized sample (n = 244)
were weighed into small tinfoil containers using an analytical balance (Sartorius Micro Pro 11, Göttingen, Ger-
many). 60 mg of tungsten$^{VI}$oxide catalyst was added and the tinfoil tightly sealed. This sealed container was
placed into the Elemental analyser (El Cube, Elementar Analysensysteme, Hanau, Germany) for subsequent
determination of total carbon (C$_t$), total nitrogen (N) and total sulfur (S) concentrations. To determine total or-
ganic carbon (C$_{org}$), another aliquot was treated with a drop of 10 % HCl solution. After gas release (CO$_2$ ↑), the
sample was treated as before. Resulting C-concentration equals the amount of C$_{org}$; prerequisite for calculating
C$_{org}$/N ratios. Calibration and daily factor determination were done using 4-aminobenzenesulfonic acid. Certified
reference material (ORIS, BHA-1) and our in-house reference material for tropical soil (BraSol) were added as
unknown samples and treated accordingly. Quantification limits were C: 0.040, N: 0.003, S: 0.003 wt-%). Sam-
ple duplicates reproduced identical values for carbon (RSD ≤1 %); standard deviations remained under 5% RSD
for nitrogen and sulfur. Reference material concentrations were always reproduced within their recommended
ranges. See Matschullat et al. (2018) for more details.
**2.5    Statistical evaluation**
Non-parametric techniques were used to calculate basic descriptive statistics (Hall and Sheather, 1988; Sheather
and McKean, 1987). Graphics were drawn with the program R (3.4.3) using the package *tidyverse*, including
*ggplot 2*.
**3    Results and discussion**
**3.1    pH-values and electrical conductivity**



TOP soil pH-values were acidic to strongly acidic with median lows of 4.2 in the dry season ($pH_{H2O}$; Table 1).
Average BOT $pH_{H2O}$ values of 4.3–4.6 in forest-covered soil; 4.6–4.9 in post-forest soil point at reduced weather-
ing rates in lower soil horizons. Forest soils were more acidic throughout in both TOP and BOT levels ($\Delta \geq 0.3$
$pH_{H2O}$ units) as compared to post-forest soils. This is likely due to much stronger rhizosphere metabolism in for-
ests and considerably lower biomass in post-forest soils (Jones et al., 2003). Considerable cation exchange ca-
pacity throughout can be derived from the delta pH between its determination in water and in $CaCl_2$ (Mekaru and
Uehara, 1972; Sanchez, 2019: 203; Table 1). Individual liming of agricultural soil cannot be excluded, resulting
in higher pH-values at those sites (Figures 2, 3). Independent of land cover, intra-annual dynamics are high with
a $pH_{H2O}$ delta of up to 0.7 units or 63 $\mu$mol $H^+$ $L^{-1}$ between (extreme) dry season and wet season (TOP); weaker in
BOT material ($\Delta$ 0.3 $pH_{H2O}$ units; Table 1; Figure 2). Highly reduced hydrolysis during the dry season liberates
less buffering cations (Ca, Mg) from the upper soil layers; soil solution pH drops. This effect is even noticeable
in $pH_{CaCl2}$ determinations, yet more subtle. Intra-annual soil pH swings should influence nutrient availability and
microbial community dynamics. Relevant differences emerge between the central and southern parts of Amazo-
nas state with considerably higher pH-values in TOP soil material after deforestation in the south (Figure 2).
This change signal is visible in BOT soil, too, yet not as pronounced.
Soil solution electrical conductivity ($\lambda$ in $\mu$S $cm^{-1}$) serves as proxy for solute transport in water-filled pore space
(WFPS). $\lambda$   was significantly higher in the dry season as compared to the wet season, and forest soil showed
consistently higher ⊁than post-forest soil (Figure 3), corroborating the pH-value results. BOT material presents
even lower soil solution electrical conductivities (27–34 $\mu$S $cm^{-1}$ in forest soil; 17–26 $\mu$S $cm^{-1}$ in post-forest soil;
ranges represent intra-annual variation). The median maximum difference between dry and wet season was 50
$\mu$S $cm^{-1}$ (Table 1; Figure 2).
Decreasing soil pH with increasing electrical conductivities in the dry season are clearly noticeable. Yet, their
effect on soil fertility and element dynamics may be too small to be of practical relevance at any given site. $\lambda$ is
considerably more dynamic in forest soil as compared to post-forest soil (Table 1, Figure 3 and electronic sup-
plement Table A02). This difference may result from soil physical changes such as decreasing porosity and in-
creasing density after deforestation.
Materials dominantly displayed light yellowish brown (10YR 6/4) colour characteristics (electronic supplement
Table A02), indicative of Fe oxihydroxides (hematite, goethite). Typical humic A-horizons (Ah) were generally
very thin. The deeper mineral soil layer (BOT) always showed slightly lighter and more reddish colours (very
pale brown, 10YR 7/4; indicating prevalence of goethite) with significantly different chemical composition from
TOP soil material (Tables 1–4).
**3.2     ORG material carbon, nitrogen and sulfur (CNS)**
Soil litter (ORG) accumulation in the Amazon basin changes radically between dry and wet seasons ($\Delta$ of up to
40 cm). We did not perform quantitative measurements, yet data from Sanches et al., (2008) and Valentini et al.,
(2008) corroborate this observation with intra-annual variation factors of 3 to 4. Due to an unusually strong EN-
SO event from 2015 to 2017 (Jiménez-Muñoz et al., 2016), seasonal differences were particularly pronounced
between our sampling campaigns Ph_02 and Ph_03. Litter accumulation varied of a few centimeters (Ju-



ly/August) to several decimeters thickness (February/ March). Litter material always was much thicker under
forest canopy as compared to any post-forest land cover.
The litter layer showed median concentrations between 34 and 49 wt.-% ($C_t$), 27 to 46 wt.-% ($C_{org}$), 0.6 to 2.3 wt.-
% (N), 0.09 to 0.23 wt.-% (S) and $C_{org}$/N ratios between 18 and 65 (electronic supplement Table A02). A more
differentiated view is needed to see regional and temporal differences. Significant difference emerged for CNS
with higher values in the central and rainier part (Table 2). Independent of geographical and morphological posi-
tion, forest litter shows lower C and N values than post-forest material. In the central part, post-forest land cover
presents higher C and N concentrations than forest soil litter, while lower ones were determined in post-forest
litter in the south. Intra-annual relative differences of ±5 % from the median value occur, with a clear tendency
to higher values in the rainy season; again independent of land use (electronic supplement Table A02). Forest
litter clearly shows faster decomposition with much lower $C_{org}$/N ratios as compared to post-forest soils. This is
particularly strong in the southern (dryer) part and additionally supported by the comparison between wet and
dry season (Table 2). This seasonal shift is accompanied by CNS losses that are particularly high for N (about -
20 %). In general, losses are higher in post-forest material (Table 4).

### 3.3 Mineral soil carbon, nitrogen and sulfur

Carbon (total and organic), nitrogen and sulfur were fully quantifiable in all samples. A non-differentiated com-
parison (central vs. southern part, forest vs. post-forest sites) of the new ERA data with global (UCC, WSA,
WISE30sec) and European (FOREGS, GEMAS) average data (Table 3 for references) show that upper crustal
data (UCC) are inappropriate for comparison, that world soil average (WSA) sulfur data lie far above both Euro-
pean soil data (FOREGS, GEMAS) and the Amazon basin soils, while the new ferralsol data ($FR_x$) from the
WISE30sec approach (Batjes, 2016) show similar values for $C_{org}$ and N (Table 3). European soil CNS averages
(FOREGS, GEMAS) are in very good agreement with the new Amazonas data.
Intra-annual changes emerge between both land-cover types with generally stronger losses in post-forest soils,
when comparing the two extreme seasons Ph_02 and Ph_03 (Table 4). Inverse conditions emerge for litter mate-
rial (ORG) as compared to the mineral soil. ORG showed lower concentrations throughout in the extreme dry
season (Ph_02) as compared to the wet season (Ph_03), while TOP and BOT material mostly showed no change
or slightly decreasing (ø -5 to -20 %) concentrations from the extreme dry to the wet season (Table 4). Only TOP
soil in the central part of the basin showed increasing concentrations between the seasons (ø +5 to +15 %).
Related results for sulfur are non-conclusive. The C/S ratio is normal (about 200 in TOP; Blume et al., 2016:
ff.). While seasonal changes clearly occur (Table 3), they show partly contradictory behavior in the three
different materials. The dominant oxisols in upland (terra firme) environments of the Amazon basin yield above-
average amounts of clay minerals (median 42%; electronic supplement Table A02) and yet show very good
drainage characteristics under forest land-cover (Table 1 with λ as proxy for drainage). Under oxidizing condi-
tions, available dissolved sulfur (as sulfate) would likely bond to abundant Fe in the system, forming ferrous
sulfate and not wash out.
**TOP mineral soil (0–20 cm).** Although Sanchez (1977) reported 2.01 wt-% of organic carbon in Brazilian
oxisols, corroborated by work from Sanchez (2019: 264), the misleading notion that tropical soils are generally
nutrient poor and show low CNS values is still widespread. Organic carbon dominated the C species (median 81



% for all data; Table 2). TOP material C is likely younger (years) than C in lower soil parts (decades to centuries
in BOT and beyond; Cuevas, 2001; Schmidt et al., 2011; Trumbore and Carmargo, 2009). The south of the basin
presented lower total $C_{org}$ and N values than its central part, and forest soils generally show higher values than
post-forest soils, independent of geographical position (Figures 4 and 5). This is less pronounced to invisible in
the south, likely reflecting an adaptation to earlier deforestation. Major deforestation started only after the con-
struction of the Transamazônica Highway (since 1972) with significant migration and settlement throughout the
basin (Fearnside, 2005). Therefore, more pronounced alterations in the C soil balance should have occurred only
within the last ±50 years. Related differences are very visible in the central part with losses of $C_t$ (-13%), $C_{org}$ (-
15%) and $N_t$ (-12%), whereas the southern parts do not show any significant difference between TOP soil CNS
concentrations under forest canopy or post-forest land use (Table 4). We assume that this is due to the fact that
the south faces much more human impact and remaining forest stretches are influenced by edge effects (Barros
and Fearnside, 2016). This hypothesis is corroborated by our data from Apuí (Loc. 07, 08) and Lábrea (Loc. 10)
for more natural conditions versus those from Humaitá (Loc 09) and Boca do Acre (Loc 13) with more disturbed
conditions. Observed seasonal dynamics are much stronger at the more natural-condition sites (electronic sup-
plement Table A02).
While CNS decreased from wet to dry season in organic litter material (= biodegradation and export), TOP min-
eral soil shows differentiated behaviour. $C_t$ and $C_{org}$ increased in forest soil, and $C_{org}$ decreased in post-forest soil,
pointing at slower biogeochemical turnover in post-forest soil with reduced microbial activity and less pore
space. Sinking pH-values likely inhibits microbial activity, too, enhancing possibly drought inhibition (Barbhui-
ya et al., 2004). Nitrogen decreased under both soil cover types, albeit stronger in post-forest soil. This observa-
tion has been made in both the central parts of the basin and in its south. In the south, differences between for-
ested (large increase) and post-forest soils (moderate increase to decrease) corroborate the hypothesis of changes
in biogeochemical metabolism between forested and post-forest soils (Table 4). It is very likely that plants, par-
ticularly trees take up the elements and soil serves as temporary storage.
**BOT mineral soil (30–50 cm).** Not surprisingly, $C_{org}$ and N data show considerably lower concentrations in the
BOT layer as compared to TOP material (Figures 4 and 5). Jobbágy and Jackson (2000) calculated the relative
SOC amounts by depth. Their compilation resulted in roughly a factor of 2 between the layers equivalent to our
TOP and BOT samples; similar to this work. CNS are lower in BOT than TOP and increase from wet to dry
season under forest canopy, with much smaller increase values for post-forest soils (Table 4). It is noteworthy
that these seasonal dynamics obviously influence the entire rhizosphere and are not restricted to the uppermost
material with the highest root mass. The smaller increase in post-forest soil can be explained by considerably
longer turnover rates due to relatively reduced soil biological metabolism (Rangel-Vasconcelos et al., 2015).
This is corroborated by slightly lower $C_{org}/N$ ratios in the $BOT_n$ materials (electronic supplement Table A02).
Given a time-lag maximum of several decades after deforestation only, distinctly lower values (ca.
-20 %) for C and N in post-forest soil is quite a strong signal (Table 2). No significant change emerged for sulfur
between forested and deforested soil, contradictory to results from McClung et al. (1959; cited in Sanchez,
2019), which may not apply due to their Cerrado-based approach. Our median values of all forested versus all
post-forest sites deliver clear results, whereas individual sites may not always show conclusive results between
dry and wet season samples (electronic supplement Table A02).





This comparison of averages (all forest, all post-forest sites) hides that the central and southern parts of the re-
gion behave differently, with the southern sites showing lower values for all four components, independent of
land cover (Table 4, Figures 4 and 5). Yet, there is the distinct difference between forest and post-forest soil. In
the central (wetter) part of the basin, decreases of -22 % $C_t$ and -13 % $C_{org}$, -13 % N occurred. No significant
change is visible in BOT material from the southern parts of the basin.
**3.4    Upscaling**
Based on ORG, TOP and BOT concentration data, material densities and estimated areas, we attempt upscaling
our CNS results for one hectare of terra firme in the Amazon basin (Table 5; Figure 6).
Assuming a conservative annual average litter (ORG) thickness of 5 cm under forest canopy and of 1 cm at post-
forest sites, and an average litter density of 0.03 g cm³ (Chojnacky et al., 2009), one hectare would generate a
mass of 15,000 kg under forest canopy and 3000 kg on post-forest soil. Based on average CNS concentrations,
pools would yield 6150/1260 kg of total carbon, 5550/1170 kg of $C_{org}$, 240/42 kg of N, and 26/5 kg of S in for-
est/post-forest environments, respectively (Table 5; Figure 6). Obviously, already visible losses are in the order
of magnitude of 60 % between forest and post-forest soil.
With an average TOP thickness of 20 cm (to stay in line with the sampling depth) and an average density of 1.3
g cm³ (Neves et al., 2003) each hectare of upland soil delivers about 2600 metric tons with pools of about 60 tons
of $C_t$ (46 tons $C_{org}$) in forest and 46 (40) tons in post-forest environments, 4.6/3.6 tons of N and 0.8/0.6 tons of S
(Figure 6). The soils are generally highly acidic (Table 1). Higher seepage water availability in the wet season
obviously discharges mobilizable cations, leading to lower conductivities, while these cations sorb onto soil
particles in the dry season.
Using a similar mass-balance approach as for the TOP, a 20 cm-thick lamina of BOT material with likely slight-
ly higher density (1.4 g cm³), producing a total mass of 2800 tons per hectare, would yield pools of about 31 tons
of $C_t$ (25 tons of $C_{org}$), 2.2 tons of N and 0.6 tons of S in forested soil. Under post-forest conditions, values are 25 t
$C_t$, 22 t $C_{org}$, 2.2 t N, and 0.6 t S (Figure 6). This is expectedly lower than in TOP soil, yet shows the relevance of
deeper C dynamics (Jobbagy and Jackson, 2000; Schmidt et al., 2011).
Most root mass resides in the top 50 cm of the local soil profiles (Canadell et al., 1996; Schenk and Jackson,
2002; Sternberg et al., 1998; our own observations). Therefore, the combined TOP and BOT samples (with BOT
calculated here to 30 cm thickness) should represent the bulk rhizosphere. Figure 6 shows the pool sizes for one
hectare in comparison between forested and post-forested soil. The CNS pools are relatively high with about 110
t of $C_t$, 90 t of $C_{org}$, 8 t of N, and 2 t of S per hectare forest soil. Pools in post-forest soil are roughly 20 % lower.
Given that such loss occurred within a maximum time of 50 years since deforestation (at some sites, deforesta-
tion happened just a short time ago), similar effect must be expected in other parts of the Amazon basin that are
under deforestation risk in the near future.
**4    Conclusions**
The initially formulated key questions can be answered as follows:





• What are CNS concentrations in Amazon basin soils – and how do these relate to other world soils?
Surprisingly, average CNS concentration (in wt-% TOP$_{Road}$: C$_t$ 2.2, C$_{org}$ 1.8, N 0.17, S 0.03) are similar to those
of Europe with Mediterranean to sub-polar climatic conditions (mostly temperate; in wt-% TOP FOREGS: C$_t$
2.2, C$_{org}$ 1.7, N 0.17, S 0.02). The range of Amazon basin CNS values appears to be considerably more
narrow; logical given more homogenous climatological and lithological conditions.
• Are there differences in nutrient status and carbon between forested and post-forest land? Differentiation is
needed between the central and southern parts of the basin, and between the upper (TOP) and lower (BOT)
soil layers within the rhizosphere. C$_t$ and C$_{org}$ are relatively enriched in central Amazonas forest soil (TOP and
BOT), while depleted in litter material (ORG) in comparison with post-forest soil (Table 2). In the southern
parts of the basin, no conclusive or significant difference emerges for CNS in mineral soil, while post-forest
soils are slightly enriched in the ORG layer. Such difference between the more wet and relatively less
disturbed sites in the central part and the south suggest that edge effects already dominate the southern
locations. The other, most likely influential driver is the underlying lithology with younger sedimentary
material in the central part but much older and metamorphic and magmatic materials in the southern part.
• Is the CNS status similar throughout the year or do we see intra-annual dynamics? Such dynamics are visible,
albeit subtle. Between wet (Phase 03) and dry season (Phase 02) as the two extremes in our study, ORG
material shows losses throughout; on average more so at post-forest sites. Gains emerged generally in
mineral soil with mostly stronger signals in forest than post-forest soil (Table 4). Again, significant
differences emerged between the central and southern parts of the basin with stronger gains in the south in
forest TOP soil. A strong gain in total S appeared in BOT of the central part as an anomaly (+58%). Such
intra-annual dynamics, very likely soil water-dependent, suggest significant influence of future hydrological
changes in the basin, as suggested by various authors (Lenton et al., 2008).
• How do mineral soil CNS concentrations relate to the organic litter layer? On average, ORG material shows
higher concentrations of C$_t$ and C$_{org}$, of N and of S than mineral soil (Table 2). A distinction emerged between
the central and southern parts. The highest ratios occur for C$_t$ and C$_{org}$. In the central part, these ratios are
lower for forest sites than for post-forest ones, while there is no significant difference in the southern part.
This finding may corroborate the previous hypothesis that edge effects already dominate soil chemistry in
this sub region.
• Is the hydrological difference between the more humid central part of the basin and its southern "shoulder"
relevant? The central parts appear to be better buffered against change as compared to the southern part,
which already is significantly more dry. Since deforestation in the south is partly extreme (exceeding 90 %),
such evidence suggests that increasing drought conditions will exacerbate the situation, leading to even more
drought in the dry season.
• Which consequences may future increasing drought conditions have for CNS behaviour in Amazon basin
soils? Given our observation of considerably lower average pH$_{H2O}$ values and higher electrical conductivity in
the dry season as compared to wet seasons, it can be inferred that the dry-season accumulation of (organic)
acids with related liberation of easily dissolvable cations will increase chemical erosion and subsequent
nutrient loss. This observation is more pronounced in forest soil as compared to post-forest soil, likely an
effect of reduced porosity and water conductivity in post-forest soils. Given the initial nutrient scarcity of
inner humid-tropical soils, increased loss is certainly unwanted.





**Acknowledgements**. This work is part of the bi-national project EcoRespira-Amazon (http://blogs.hrz.tu-
freiberg.de/ecorespira/) and has been generously supported by the German-Brazilian cooperation programme
(DAAD 57204144). On the German side, the Federal Ministry for Economic Cooperation and Development
(BMZ) with Deutsche Gesellschaft für Internationale Zusammenarbeit (GIZ) and the German Academic Ex-
change Service (DAAD) financed and organized the project under the umbrella of Nova Parcerias (NoPa2) since
2015. The Federal Ministry for Science, Technology, Innovation and Communication (MCT), the Federal Minis-
try of Education (MEC) and the Coordenação de Aperfeiçoamento de Pessoal de Nível Superior (CAPES) were
responsible on the Brazilian side.
The authors are most thankful for this financial support, without which this and several other studies within the
project would not have been possible. We also wish to acknowledge infrastructural and technical support from
Embrapa in Manaus and Porto Velho as well as the hands-on help from Ednilson Alves Figueiredo, Joel Gomes,
Marcelo Renan de Oliveira Teles, Cintia R. Souza (all Embrapa Amazônia Ocidental) and Kikue Moroya from
IPAAM. Last but not least, our appreciation goes to Kamal Zurba and Thomas Drauschke for their dedicated
field work (manual soil drilling in Amazon Basin soils is no fun), to Karsten Gustav (pH, EC, soil colour) and
Richard Hammig (TPI) for their Bachelor thesis work, and to the Freiberg laboratories' team, namely Dr. Alex-
ander Plessow with Elvira Rüdiger (EA), for excellent analytical work and lab quality control. Thank you ALL.

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



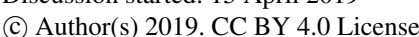

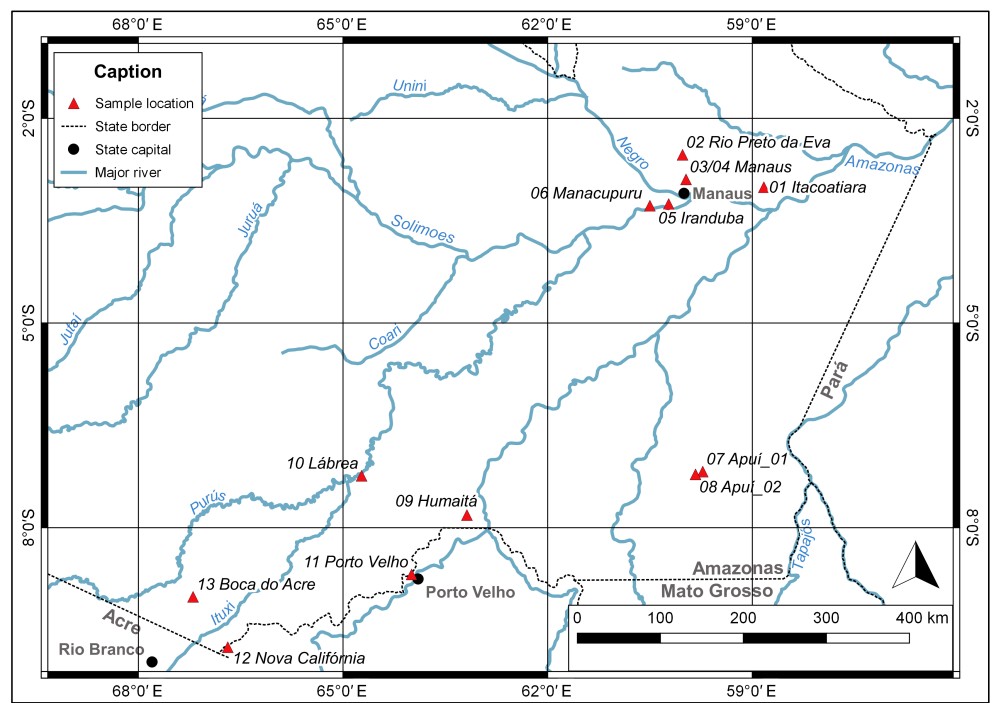

**Figure 1. Location map Amazon basin (based on QGIS: 2.18.14 Las Palmas)**

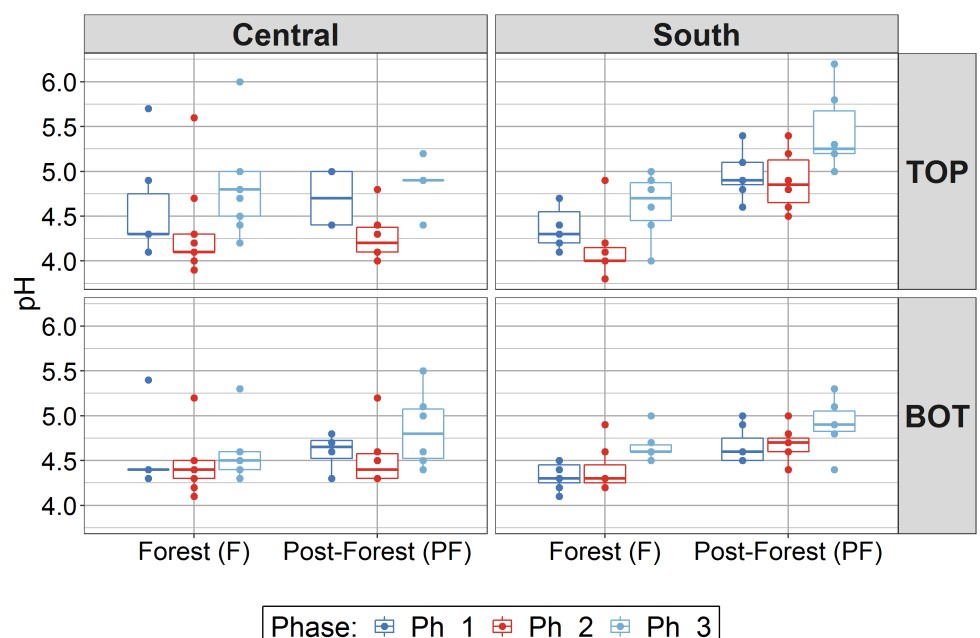

**Figure 2. Boxplots show seasonal $pH_{H2O}$ dynamics in Amazonas upland soils, differentiated by region (central and**
**south) as well as soil depth (TOP, BOT). See text for details**



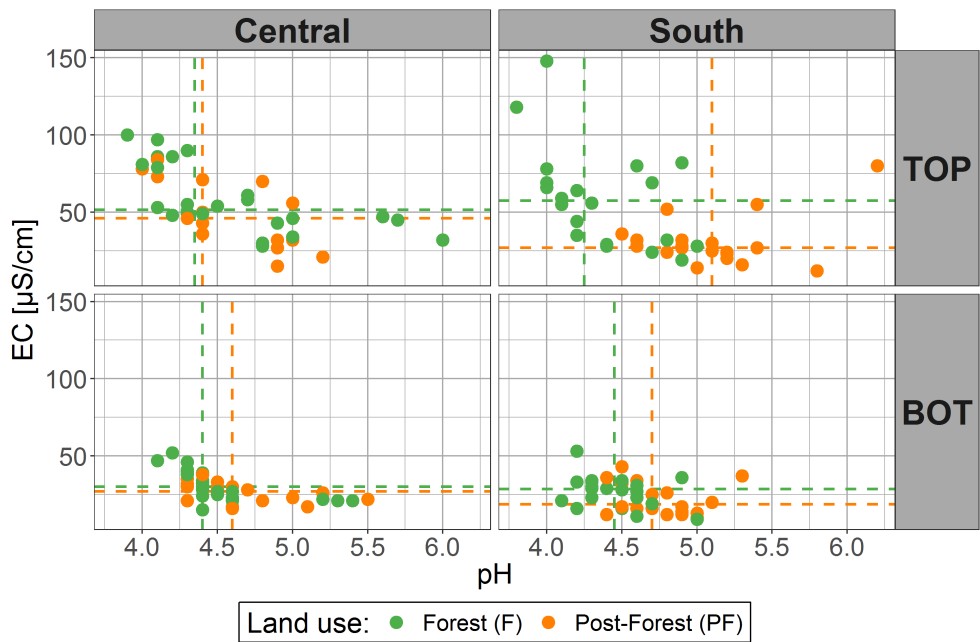


**Figure 3. Soil solution electrical conductivity (◆) versus pH-value (pH$_{H2O}$) in TOP and BOT samples from all ERA sites, differentiated by the central and southern locations and by forest (F) and post-forest (PF) land use**

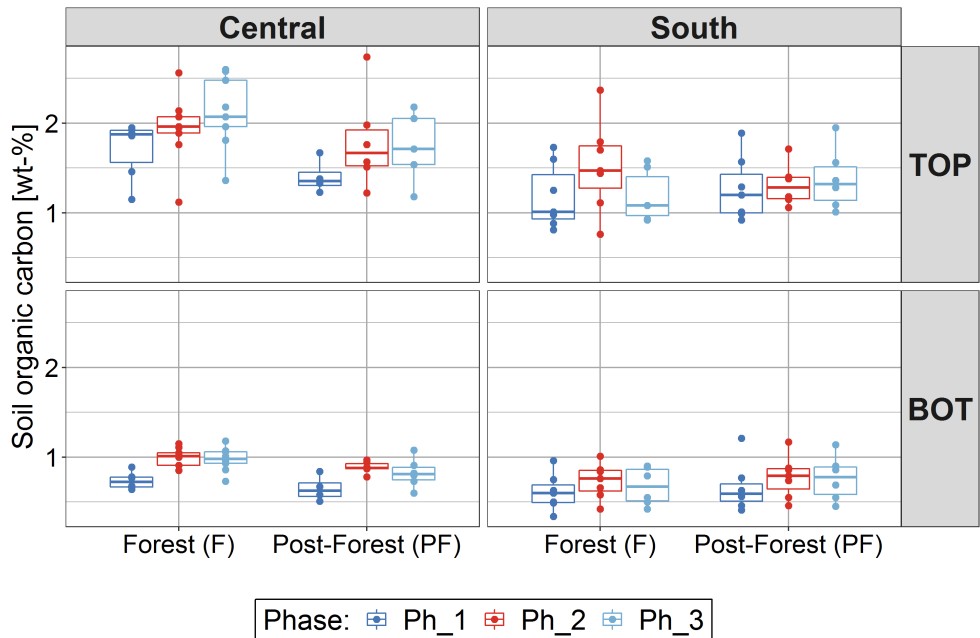


**Figure 4. Soil organic carbon boxplots show seasonal dynamics relative to Amazonas region (central and south) as well as soil depth (TOP, BOT). See text for details**






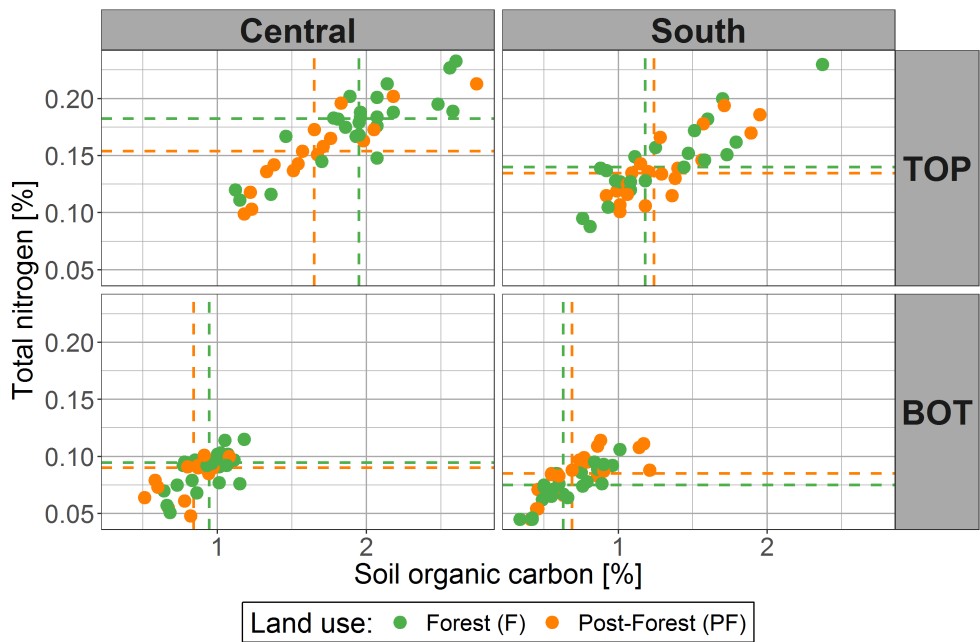

**Figure 5.** Total soil organic carbon and nitrogen (wt-%) in TOP and BOT samples from all ERA sites, differentiated by the central and southern locations and by forest (F) and post-forest (PF) land use

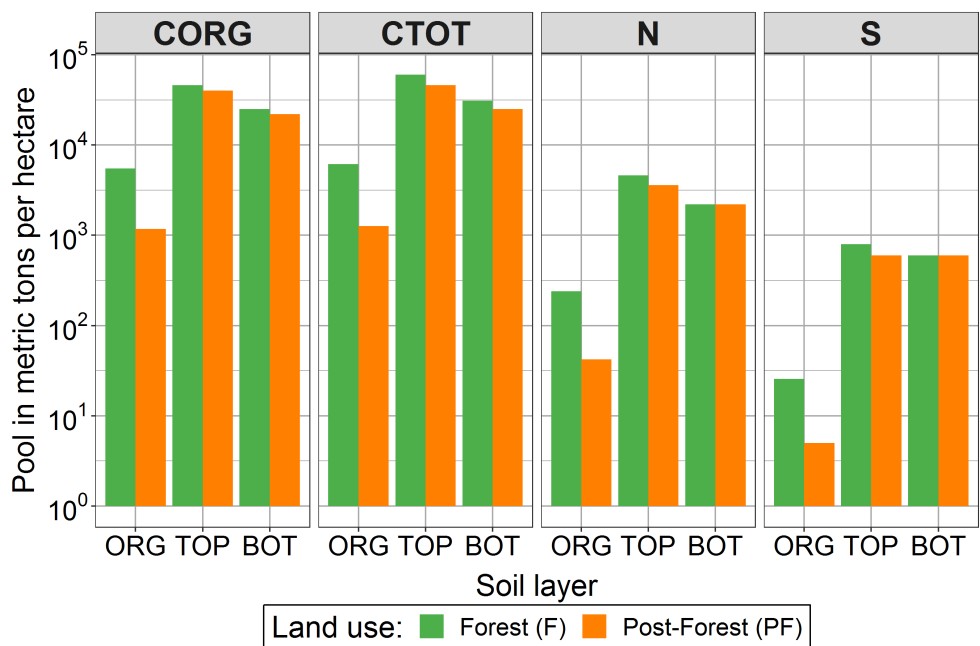

**Figure 6.** Average total and organic carbon ($C_t$, $C_{org}$), nitrogen (N) and sulfur (S) pool sizes per hectare of Amazon Basin soil layers under forest and post-forest land cover

551

(c) Author(s) 2019. CC BY 4.0 License.



**Table 1. Median\* soil pH (pH$_{H2O}$ and pH$_{CaCl2}$) and electrical conductivities (✦in $\mu S$ cm$^{-1}$) in three subsequent field campaigns, representing moderately wet (Ph_01), very dry (Ph_02) and very wet (Ph_03) seasons. Data are differentiated by land cover (Forest and Post-Forest)**

|  | pH$_{H2O}$ | | | pH$_{CaCl2}$ | | | λ (μS cm$^{-1}$) | | |
|---|---|---|---|---|---|---|---|---|---|
| Season | Ph_01 | Ph_02 | Ph_03 | Ph_01 | Ph_02 | Ph_03 | Ph_01 | Ph_02 | Ph_03 |
| TOP$_{total}$ | 4.6 | 4.2 | 4.9 | 3.9 | 3.8 | 3.9 | 43 | 69 | 32 |
| TOP$_{Forest}$ | 4.3 | 4.1 | 4.8 | 3.8 | 3.7 | 3.8 | 49 | 81 | 32 |
| TOP$_{Post-Forest}$ | 4.9 | 4.6 | 5.2 | 4.0 | 4.0 | 4.0 | 32 | 46 | 23 |
| BOT$_{total}$ | 4.5 | 4.5 | 4.6 | 3.9 | 3.9 | 4.0 | 24 | 30 | 22 |
| BOT$_{Forest}$ | 4.4 | 4.3 | 4.6 | 3.8 | 3.9 | 4.0 | 28 | 33 | 23 |
| BOT$_{Post-Forest}$ | 4.6 | 4.6 | 4.9 | 3.9 | 3.9 | 4.0 | 23 | 25 | 17 |

\*Median: average of all median values from all data sub collectives

**Table 2. Median CNS data (wt-%) for all samples (n = 85), differentiated by geographical position and land cover**

| Group | Type | C$_t$ | C$_{org}$ | N | S | C$_{org}$/N |
|---|---|---|---|---|---|---|
| all data | ORG | 42 | 38 | 1.4 | 0.17 | 27 |
|  | TOP | 1.9 | 1.6 | 0.16 | 0.03 | 10 |
|  | BOT | 1.0 | 0.8 | 0.08 | 0.02 | 9 |
| all central | ORG | 43 | 39 | 1.6 | 0.19 | 27 |
|  | TOP | 2.4 | 2.0 | 0.18 | 0.03 | 11 |
|  | BOT | 1.1 | 0.9 | 0.09 | 0.03 | 10 |
| all south | ORG | 41 | 37 | 1.4 | 0.16 | 30 |
|  | TOP | 1.5 | 1.2 | 0.14 | 0.02 | 9 |
|  | BOT | 0.8 | 0.7 | 0.08 | 0.02 | 9 |
| all forest | ORG | 41 | 37 | 1.6 | 0.17 | 26 |
|  | TOP | 2.3 | 1.8 | 0.17 | 0.03 | 10 |
|  | BOT | 1.1 | 0.9 | 0.08 | 0.02 | 10 |
| forest central | ORG | 41 | 29 | 1.6 | 0.17 | 27 |
|  | TOP | 2.4 | 2.0 | 0.18 | 0.03 | 10 |
|  | BOT | 1.1 | 0.9 | 0.09 | 0.03 | 10 |
| forest south | ORG | 41 | 38 | 1.8 | 0.17 | 25 |
|  | TOP | 1.5 | 1.2 | 0.14 | 0.02 | 9 |
|  | BOT | 0.8 | 0.7 | 0.08 | 0.01 | 9 |
| all post-forest | ORG | 42 | 39 | 1.4 | 0.16 | 32 |
|  | TOP | 1.8 | 1.5 | 0.14 | 0.02 | 11 |
|  | BOT | 0.9 | 0.8 | 0.08 | 0.02 | 9 |
| post-forest C | ORG | 45 | 42 | 1.8 | 0.21 | 26 |
|  | TOP | 2.1 | 1.7 | 0.16 | 0.03 | 11 |
|  | BOT | 1.0 | 0.8 | 0.08 | 0.02 | 10 |
| post-forest S | ORG | 41 | 37 | 1.2 | 0.14 | 32 |
|  | TOP | 1.5 | 1.2 | 0.13 | 0.02 | 9 |
|  | BOT | 0.9 | 0.7 | 0.08 | 0.02 | 8 |

Data rounded for clarity. Detailed values (all meaningful digits) in electronic supplement





**Table 3. Median values for carbon (C$_t$ and C$_{org}$), total nitrogen (N) and total sulfur (S) in upper continental crust**
**(UCC), World Soil Averages (WSA), global ferralsols (WISE30sec), in European TOP and BOT mineral soil**
**(FOREGS and GEMAS) and in Amazon basin upland (terra firme) ferralsols (TOP, BOT) with forest (F) and post-**
**forest (PF) land cover**

| (wt-%) | C$_t$ | C$_{org}$ | N | S |
|---|---|---|---|---|
| UCC | 0.32 | n.d. | 0.008 | 0.06 |
| WSA | n.d. | n.d. | n.d. | 0.08 |
| WISE30sec TOP | n.d. | 1.7 | 0.15 | n.d. |
| WISE30sec BOT | n.d. | 1.0 | 0.10 | n.d. |
| FOREGS TOP | 2.2[#] | 1.7 | 0.17[#] | 0.02 |
| FOREGS BOT | n.d. | 0.4 | n.d. | 0.01 |
| F-TOP | 2.2 | 1.8 | 0.17 | 0.03 |
| F-BOT | 1.0 | 0.9 | 0.08 | 0.02 |
| PF-TOP | 1.8 | 1.4 | 0.14 | 0.02 |
| PF-BOT | 0.9 | 0.8 | 0.08 | 0.02 |

All values rounded for clarity. "n.d.": not determined. "F": forest, "PF": post-forest. UCC data: Rudnick and Gao
(2014). WSA and total carbon in UCC: Reimann and de Caritat (1998); WISE30sec data for global ferralsols, TOP
= 0–20 cm, BOT = 20–40 cm: Batjes (2016) and https://www.isric.org/explore/wise-databases; FOREGS data:
Salminen (2005), #C and N data: Matschullat et al. (2018), based on the GEMAS project (Reimann et al., 2014)
**Table 4. Wet season – dry season dynamics in CNS data (relative percentage change), differentiated by geographical**
**position (C = central, S = south) and land cover (forest and post-forest)**

| | | Median values all sites in wt.-%* | | | Wet to dry season | |
|---|---|---|---|---|---|---|
| | component | Ph_01* | Ph_02 | Ph_03 | forest | post-forest |
| ORG-C | C$_t$ | 41.6 | 41.7 | 46.6 | - 9% | - 13% |
| | N | 1.35 | 1.49 | 1.80 | - 21% | - 17% |
| | S | 0.201 | 0.172 | 0.194 | - 8% | - 14% |
| ORG-S | C$_t$ | 40.6 | 40.8 | 43.3 | - 10% | - 5% |
| | N | 1.38 | 1.07 | 1.57 | - 18% | - 29% |
| | S | 0.158 | 0.129 | 0.169 | - 9% | - 27% |
| TOP-C | C$_t$ | 2.12 | 2.50 | 2.31 | + 10% | + 3% |
| | C$_{org}$ | 1.47 | 1.95 | 2.05 | + 5% | - 6% |
| | N | 0.159 | 0.165 | 0.184 | - 2% | - 4% |
| | S | 0.032 | 0.029 | 0.030 | - 12% | + 4% |
| TOP-S | C$_t$ | 1.52 | 1.81 | 1.45 | + 35% | + 7% |
| | C$_{org}$ | 1.10 | 1.40 | 1.18 | + 36% | - 3% |
| | N | 0.135 | 0.143 | 0.136 | + 15% | - 4% |
| | S | 0.023 | 0.021 | 0.021 | + 5% | - 8% |
| BOT-C | C$_t$ | 0.86 | 1.11 | 1.05 | + 10% | ± 0% |
| | C$_{org}$ | 0.68 | 0.94 | 0.95 | + 3% | + 8% |
| | N | 0.075 | 0.093 | 0.092 | + 5% | + 8% |
| | S | 0.019 | 0.036 | 0.028 | + 12% | + 58% |
| BOT-S | C$_t$ | 0.83 | 0.99 | 0.87 | + 13% | + 6% |
| | C$_{org}$ | 0.60 | 0.78 | 0.67 | + 13% | + 1% |
| | N | 0.073 | 0.085 | 0.081 | + 13% | + 8% |
| | S | 0.015 | 0.018 | 0.016 | + 24% | ± 0% |

*The individual campaigns returned with slightly different numbers of samples: Ph_01: ORG-C 9,
ORG-S 14; TOP-C 10, TOP-S 14; BOT-C 10, BOT-S 14; Ph_02: ORG-C 14, ORG-S 13; TOP-C 15, TOP-S
13; BOT-C 15, BOT-S 14; Ph_03: ORG-C 15, ORG-S 10; TOP-C 15, TOP-S 12; BOT-C 15, BOT-S 12
Phase 01 was excluded from the intra-annual calculation, since it was an untypical, ENSO-driven,
weak wet season



**Table 5. Nutrient element stocks (in kilograms) in the three pools of litter (ORG), mineral topsoil (TOP: 0–20 cm) and**
**deeper mineral soil (BOT: 20–50 cm), differentiated by forest and post-forest land cover. Values present total stocks**
**per hectare, calculated from the median data in Tables 2–4**

|  | ORG | | TOP | | BOT | |
|---|---|---|---|---|---|---|
|  | **Forest** | **Post-Forest** | **Forest** | **Post-Forest** | **Forest** | **Post-Forest** |
| $C_t$ | 6150 | 1260 | 60000 | 46000 | 31000 | 25000 |
| $C_{org}$ | 5550 | 1170 | 46000 | 40000 | 25000 | 22000 |
| N | 240 | 42 | 4600 | 3600 | 2200 | 2200 |
| S | 25.5 | 5 | 800 | 600 | 600 | 600 |

577                                                     All values rounded for clarity





**Electronic supplement**
**Table A01**. Average values for temperature and accumulated precipitation versus the climate normal
1961–1990 (CN_6190); based on eight INMET meteorological stations in Amazonas

| Maximum temperature (°C) | | | | | | | |
|---|---|---|---|---|---|---|---|
| **FEB** | Barcelos | Fonte Boa | Itacoatiara | Lábrea | Manaus | Manicoré | S.G. Cach. (Uaupés) | Tefé |
| 2016 | 34.7 | 32.6 | 31.7 | | | 33.3 | 34.0 | 33.7 |
| 2017 | 34.0 | 31.1 | 30.5 | 32.0 | 31.0 | 31.5 | 33.9 | 31.7 |
| CN_6190 | **31.9** | **31.1** | **30.5** | **29.1** | **30.4** | **30.5** | **31.0** | **31.2** |
| **MAR** | Barcelos | Fonte Boa | Itacoatiara | Lábrea | Manaus | Manicoré | S.G. Cach. (Uaupés) | Tefé |
| 2016 | 33.1 | 32.2 | 31.5 | | | 32.7 | 33.7 | 31.6 |
| 2017 | 33.8 | 31.1 | 30.8 | 32.3 | 31.4 | 32.3 | 33.4 | 31.0 |
| CN_6190 | **31.9** | **30.7** | **30.1** | **30.9** | **30.6** | **31.0** | **30.8** | **31.3** |
| **JUL** | Barcelos | Fonte Boa | Itacoatiara | Lábrea | Manaus | Manicoré | S.G. Cach. (Uaupés) | Tefé |
| 2016 | 34.0 | 31.7 | 33.1 | 34.9 | 33.4 | 35.2 | 31.8 | 32.7 |
| CN_6190 | **31.1** | **30.1** | **31.1** | **31.9** | **31.3** | **32.0** | **29.2** | **31.1** |
| **AUG** | Barcelos | Fonte Boa | Itacoatiara | Lábrea | Manaus | Manicoré | S.G. Cach. (Uaupés) | Tefé |
| 2016 | | 33.6 | 33.8 | 35.3 | 34.4 | 35.7 | 33.4 | 34.3 |
| CN_6190 | **31.6** | **31.1** | **32.0** | **32.5** | **32.6** | **33.2** | **30.3** | **31.9** |
| Minimum temperature (°C) | | | | | | | |
| **FEB** | Barcelos | Fonte Boa | Itacoatiara | Lábrea | Manaus | Manicoré | S.G. Cach. (Uaupés) | Tefé |
| 2016 | 24.6 | 22.9 | 24.7 | 23.8 | 25.4 | 24.1 | 23.2 | 24.2 |
| 2017 | 23.3 | 22.9 | 23.6 | 23.2 | 24.5 | 23.7 | 20.3 | 23.7 |
| CN-6190 | **22.1** | **22.4** | **22.0** | **20.7** | **23.1** | **22.0** | **22.6** | **22.9** |
| **MAR** | Barcelos | Fonte Boa | Itacoatiara | Lábrea | Manaus | Manicoré | S.G. Cach. (Uaupés) | Tefé |
| 2016 | 23.4 | 22.5 | 24.3 | 24.0 | 25.4 | 24.2 | 22.9 | 24.2 |
| 2017 | 23.1 | 22.8 | 23.8 | 23.4 | 24.5 | 24.0 | 20.1 | 24.0 |
| CN-6190 | **22.2** | **22.5** | **22.3** | **21.0** | **23.2** | **22.3** | **22.6** | **22.9** |
| **JUL** | Barcelos | Fonte Boa | Itacoatiara | Lábrea | Manaus | Manicoré | S.G. Cach. (Uaupés) | Tefé |
| 2016 | 23.4 | 22.4 | 23.9 | 21.1 | 25.1 | 22.9 | 20.7 | 24.0 |
| CN-6190 | **21.5** | **21.5** | **21.8** | **19.0** | **22.7** | **21.2** | **21.6** | **21.9** |
| **AUG** | Barcelos | Fonte Boa | Itacoatiara | Lábrea | Manaus | Manicoré | S.G. Cach. (Uaupés) | Tefé |
| 2016 | 23.3 | 23.0 | 24.3 | 21.8 | 25.7 | 23.3 | 20.6 | 24.0 |
| CN-6190 | **21.5** | **21.6** | **21.9** | **19.3** | **23.0** | **21.5** | **21.7** | **22.3** |
| Corrected average temperature (°C) | | | | | | | |
| **FEB** | Barcelos | Fonte Boa | Itacoatiara | Lábrea | Manaus | Manicoré | S.G. Cach. (Uaupés) | Tefé |
| 2016 | 28.2 | 27.3 | 27.9 | 28.3 | 28.2 | 27.2 | 27.3 | 27.3 |
| 2017 | 27.1 | 27.4 | 26.7 | 26.6 | 26.8 | 26.3 | 26.1 | 26.9 |
| CN-6190 | **26.3** | **25.9** | **25.8** | **25.5** | **25.9** | **25.8** | **26.1** | **26.1** |
| **MAR** | Barcelos | Fonte Boa | Itacoatiara | Lábrea | Manaus | Manicoré | S.G. Cach. (Uaupés) | Tefé |
| 2016 | 27.3 | 27.2 | 27.5 | | 27.9 | 27.1 | 27.1 | 27.1 |
| 2017 | 27.1 | 27.4 | 27.0 | 26.6 | 27.0 | 26.8 | 25.8 | 26.7 |
| CN-6190 | **26.3** | **25.5** | **25.9** | **26.0** | **26.0** | **26.1** | **26.0** | **26.2** |
| **JUL** | Barcelos | Fonte Boa | Itacoatiara | Lábrea | Manaus | Manicoré | S.G. Cach. (Uaupés) | Tefé |
| 2016 | 27.6 | 26.3 | 28.4 | 26.9 | 28.7 | 28.0 | 25.2 | 27.5 |
| CN-6190 | **25.4** | **24.8** | **26.2** | **25.0** | **26.5** | **26.1** | **24.7** | **25.8** |
| **AUG** | Barcelos | Fonte Boa | Itacoatiara | Lábrea | Manaus | Manicoré | S.G. Cach. (Uaupés) | Tefé |
| 2016 | 27.6 | 27.3 | 28.7 | 27.3 | 29.4 | 28.0 | 26.0 | 28.2 |
| CN-6190 | **25.7** | **25.4** | **26.7** | **25.5** | **27.3** | **26.7** | **25.2** | **26.3** |
| Accumulated monthly precipitation (mm) | | | | | | | |
| **FEB** | Barcelos | Fonte Boa | Itacoatiara | Lábrea | Manaus | Manicoré | S.G. Cach. (Uaupés) | Tefé |
| 2016 | 53.5 | 284 | 402.4 | 304.2 | 235.3 | 256 | 157.9 | 226.0 |
| 2017 | 162.7 | 267.1 | 377.5 | 328 | 257.4 | 390.9 | 441.2 | 317.9 |
| CN-6190 | **174.0** | **200.3** | **294.9** | **370.4** | **289.5** | **367.9** | **231.8** | **244.7** |
| **MAR** | Barcelos | Fonte Boa | Itacoatiara | Lábrea | Manaus | Manicoré | S.G. Cach. (Uaupés) | Tefé |
| 2016 | 109.8 | 175.4 | 628.9 | 295.2 | 281.9 | 391.2 | 311.8 | 408.3 |
| 2017 | 98.4 | 291.6 | 578.9 | 197 | 270 | 213.6 | 285.1 | 173.4 |
| CN-6190 | **274.2** | **265.9** | **348.0** | **371.8** | **335.4** | **293.5** | **241.7** | **301.3** |
| **JUL** | Barcelos | Fonte Boa | Itacoatiara | Lábrea | Manaus | Manicoré | S.G. Cach. (Uaupés) | Tefé |
| 2016 | 63 | 189.2 | 132.7 | 9.8 | 103.2 | 4.5 | 267.3 | 166.8 |
| CN-6190 | **198.5** | **175.7** | **129.5** | **45.7** | **85.4** | **74.7** | **247.6** | **124.5** |
| **AUG** | Barcelos | Fonte Boa | Itacoatiara | Lábrea | Manaus | Manicoré | S.G. Cach. (Uaupés) | Tefé |
| 2016 | 75 | 88.8 | 79.3 | 5.7 | 49.8 | 54.4 | 142.7 | 59.5 |
| CN-6190 | **155.8** | **153.4** | **82.8** | **83.7** | **47.3** | **75.5** | **193.7** | **101.3** |



**For Table A02, please see separate zip file**