# Peer review of "Carbon, nitrogen and sulfur (CNS) status and dynamics in"

_SOIL, 2019_

## Referee Comment (RC1) · Anonymous Referee #1 · 1 May 2019

Review of Matschullat et al paper submitted to SOIL

To answer the questions asked in this paper, the correct approach would be to pre-plan a sampling programme, based on statistical evaluation of the number of sites and samples required to produce statistically significant results. Instead the authors have chosen 11 sites in three clusters, and sampled at three places at each site, to produce their results. It is hard to accept that the locations are in any way representative, and so to draw conclusions about the Amazon terra firme soils as a whole, on the basis of the present results, cannot be justified.

A second criticism is that no statistics are given. We are told that there are seasonal

variations, and differences between forested and "post-forested" soils, but no statistical significance is provided. Only median values are given in the Tables, with no indication of the dispersion of the data. Admittedly the pH and carbon data are presented in box plots in Figures 2 and 4, but these only give qualitative impression.

Each of the above flaws is major, and they mean that the paper is unacceptable for publication. A third point that can be made is that comparing tropical soils with European soils, many of which must have been under long-term cultivation, seems illogical.

---

## Referee Comment (RC2) · Alessandro Samuel-Rosa (Referee) · 25 Jun 2019

**1   General comments**

The present manuscript describes the study of the variation of the carbon, nitrogen and sulphur contents in the soil and litter in uplands of the State of Amazonas, Brazil. The main goal of the authors was to answer long-standing research questions about the dynamic of these nutrients in the Amazon region by using more reliable data than previous studies. Unfortunately the present manuscript fall short in delivering the aimed target. Some of the reasons have already been provided in the comments of the Anonymous

[Figure]

Referee #1. In the next section I provide a few comments to address some of the key scientific issues that I identified in the present manuscript.

Before proceeding, let me mention that I think the present manuscript has the potential to provide new and interesting knowledge about the soil. It only depends on how willing the authors are to make the necessary corrections and adjustments.

**2 Specific comments**

I agree with the Anonymous Referee #1 when s/he says that the sampled data is insufficient and inappropriate to answer the initial questions and upscale the results to the entire upland Amazon basin. There is very little indication on why and how the sampling sites were selected or how representative of the entire upland Amazon basin they are. Sampling along a single year also is insufficient to make statements about the seasonal dynamics of carbon, nitrogen and sulphur contents in the soil and litter.

Interestingly, the authors have provided a very through description of the soil sampling and laboratory analysis protocols, reagents and equipment. However, aside for the little information of the procedure for selecting the sampling sites, there is no background information of upscaling methods. Of major concern is the fact that the authors completely ignored the (often) large uncertainty of upscaled results. In the same manner, as already pointed to the Anonymous Referee #1, the description of statistical methods and the presentation of results is very incipient.

It is not clear why the authors attempted to compare the soil contents of carbon, nitrogen and sulphur of the Amazon with the soil of other parts of the world. For that reason, the Anonymous Referee #1 stated that "comparing tropical soils with European soils [...] seems illogical". If the authors have any strong argument in favour of doing so, then their argument should be provided in the manuscript. But note that, when making such global comparisons, then the authors should also include other tropical forest in
the analysis as well.

Regarding the statistical analysis of the data, I want to stress that the authors should take into account the fact that the sampling sites were chosen in a non-probabilistic manner and are highly clustered. This means that geostatistical considerations could be necessary when analysing the data. Also, some sampling site are very close to major cities and/or major rivers, while other sampling sites seem to be very isolated. The authors need to demonstrate that the closeness to major rivers and cities have no (or only minor) influence on the results.

Finally, given that the present study was funded by government agencies, both Brazilian and German, the authors should present a long term data management plan. For instance, this could be done though its publication in a research data repository.

**3  Final considerations**

I conclude that the manuscript, in its current form, is not suited for publication in SOIL. I suggest the authors to make a major revision, providing more background information on how the sampling sites were selected and using more appropriate statistical analyses. Additionally, I strongly recommend the authors to avoid making any upscaling attempt with the existing data because I think that it is insufficient and inappropriate for such purpose.

---

## Author Comment (AC1) · 28 Jul 2019

Dear editor (and possibly dear referee colleague, responsible for those review comments),

We as authors are a little bit stupefied by some of the comments since they make assumptions about our work that are neither corroborated by the article itself nor by previous related publications.

Referee 1 writes: "... the correct approach would (have been) to preplan a sampling programme, based on statistical evaluation of the number of sites and samples required

to produce statistically significant results".

Our comment #01: The Amazon basin is not Central Europe or comparable in site accessibility with many other places in the world that have easy road access. So far the entire basin is represented on the Brazilian side in most publications by locations and sites near the Embrapa research sites in the states of Acre (Rio Branco), Rondonia (Porto Velho), Amazonas (Manaus) and Pará (Belém), to name the most prominent ones. In all cases, these sites are in the direct perimeter of the related capital city (maximum 30 minute drives on asphalted roads.

In order to overcome limitations that certainly do derive from those limited locations/sites, we have planned for about six months where to position locations and their sites in order to cover the Amazonas state upland (terra firme) part of the Amazon basin, acknowleding its geological, morphological, climatological and hydrological differentiation into the central (Amazon "Graben") part and the southern "shoulder" part, details of which are given in the manuscript. This selection was done using the expertise from our Embrapa team plus of the National Institute for Amazonas Studies INPA in Manaus that know Amazonas state and its soils very well. By definition, we intended to cover as large an area as possible, as far away from direct urban or municipal impact as possible, where sites represented little (if at all) disturbed rainforest. in their vicinity, we selected another site (or more) with post-forest land cover (agriculture, pastureland etc.), following more of less recent deforestation. We are rather confident that we did identify very appropriate locations to realize our study. At each sampling campaign, an Embrapa-based, highly experienced soil scientist was part of the team, confirming the choice of locations and sites. Differences (representativeness) were certainly not larger as compared to sampling e.g., only brownsoils (cambisols) of Central Europe (see FOREGS and GEMAS soil mapping projects by EUROGEOSURVEYS).

It goes without saying that an even larger number of locations and sites would be nice to have, and yet, we do not know of anyone who has undertaken with one team any related task, covering so many locations with repeated sampling and fulfilling very tight

quality control.

The referee continues: "Instead the authors have chosen eleven sites in three clusters, and sampled at three places at each site to produce their results. It is hard to accept that the locations are in any way representative, and so to draw conclusions about the Amazon terra firme soils as a whole, on th ebasis of the presented result, cannot be justified."

Our comment #02: We have not chosen eleven but 13 locations with 29 sites (some locations had more than two sites) - see figure 1. There are no three clusters. One might argue for two clusters, one being in the Amazon "graben" (locations 01 to 06) and then the rest, yet really, these are no clusters - see geological, morphological and land cover maps. The data also clearly corroborate our claim that there is no geographical bias in the data. There is, however, a clear distinction between the more wet central part and the relatively drier southern part (locations 07 to 13).

The authors partly overlap with those that successfully published a highly-cited review paper on soil respiration with in-depth discussion of representativeness (Oertel et al. 2016). Apart from the focus there on soil respiration, the vast majority of the several hundred studies that the review is based on did a) also take soil samples and b) did usually take rather fewer samples and mostly no repetitions at all.

We claim that our sampling approach is very representative indeed, fully corroborated by the data, where subsequent sampling campaigns very well reproduce the previous results. This is not only recognizable in CNS data, but also in major, minor and trace elements (65 elements of the PSE, manuscript in preparation). We can provide the evidence, if you wish.

Referee 1 writes: "A second criticism is that no statistics are given. ... only give qualitative impression".

Our comment #03: There are indeed different "schools" in earth science and soil science, some of which trust very strongly in statistics and other that focus more on field evidence and observations. It is not our place to judge between those "schools". Instead we wish to point out that Dr. Solveig Pospiech (co-author and Post-Doc with ample experience in high-level statistics) and her supervisor, Prof. Dr. Gerald van den Boogaart (highly recognized as leading specialist in geostatistics), have read all versions of the manuscript and helped with the R-scripts to perform the statistical evaluation presented in the manuscript.

Obviously, there is quite some statistics behind the figures 2 to 5 and again, very thorough quality control. Therefore, the generalized claim of referee 1 that there was no statistics given, is not true. What the referee may miss, is some tabular statistical indices such as "alphas", "r squares", etc. to describe probabilities, variances, etc. We can happily deliver that yet feel that it is not needed to fully benefit from the content of our manuscript. We do not really understand, how figures with unmistakable scales do not deliver a quantiative impression, but a "qualitative" one only.

Referee 1 ends: "Each of the above flaws is major, and they mean that the paper is unacceptable for publication. A third point that can be made is that comparing tropical soils with European soils, many of which must have been under ling-term cultivatioon, seems illogical".

Our comment #4: We do certainly not agree at all that there are such flaws. We agree that there may be a bias on our side towards field evidence and observations, while referee 1 likely prefers much stronger emphasis on explicit statistics. As mentioned above, this can be delivered, yet will not change any of the content nor of our conclusions.

The other statement is kind of strange to us, too. While not that strongly evident with this manuscript and the underlaying data, our other pedochemical data clearly show that the WSA (world soil average) data are obviously biased towards temperate climate and European and North American soils. In 2012, Patrice de Caritat (Geoscience Australia) and Clemens Reimann (Norwegian Geological Survey) published an excellent paper in EPSL, comparing European and Australian soil data and developed "Predicted Empirical Global Soil (PEGS2) reference values" (see doi 0.1016/j.epsl.2011.12.033).

Of course one can - and should - compare large-scale data sets, if the derived insight and/or message is a valuable contribution to discussion. In this particular case (topic of our manuscript), there simply exists a major bias in much of the science world towards European and North American data and their interpretation. One concrete effect is the oversimplification of tropical soils as being nutrient poor, especially in the humid tropics. Since nitrogen and sulfur are key nutrients and the organic carbon to nitrogen ratio an important indicator for soil quality, the comparison we make (and which is not even in the focus of our manuscript) is of interest to the community.

The critized "upscaling" is a truly small - and as such very clearly defined within the manuscript - attempt to show consequences of the new data, if we upscale them. The manuscript does nowhere claim that we now deliver the "truth" about Amazon basin soil CNS concentration. We upscale to one hectare, and yes, one can upscale higher and see how that relates to other assumptions already published.

We simply hope that our replies and explanations help to better understand our work, of which we are proud and certain that the data quality is of very high quality and trustworthy.

Sincerely yours, and writing for all co-authors

Jörg Matschullat

---

## Author Comment (AC2) · 28 Jul 2019

Dear Editor, dear Alessandro Samuel-Rosa as second referee,

Let me reply to your, Alessandro's comments (ASR) in the sequence of your review:

ASR 2: "I agree with the Anonymous Referee #1 when s/he says that the sampled data is in-sufficient and inappropriate to answer the initial questions and upscale the results tothe entire upland Amazon basin. There is very little indication on why and how thesampling sites were selected or how representative of the entire upland Amazon basinthey are. Sampling along a single year also is insufficient to make statements

aboutthe seasonal dynamics of carbon, nitrogen and sulphur contents in the soil and litter."

Our comments 01: Could there be a misunderstanding inasmuch as our "upscaling" exercise is of minor relevance within the context of our work and has been overemphasized by both referees? Even reading our abstract now, I have a hard time seeing that point in the limelight.

However, to question our new data as insufficient and inappropriate for answering the initial research questions demands debate: We say: 1) What are CNS concentrations in Amazon basin upland soils - and how do these compare to other world soils? We deliver exactly that with a larger number of samples as before - not modelled but truly taken (repeatedly) and measured state-of-the-art. We ask 2) if there are differences in CNS status between forest and post-forest soil - and we deliver answers. Our question to the referees: where is the misunderstanding?

The second part of the referee's comment above relates to the methodology of site selection. We can of course go into more detail - and this has partly been addressed in our replies to referee 1. Top criterion was representativity for Amazoans state upland soils. Criterion 2 was ferralsols (oxisols, latosols). Criterion 3 was acessibility within the usual contraints. This meant site acessibility in both dry and wet season. We drove (4WD) as close as possibly with our 4WD, then had to walk and carry all gear into the forest or onto farmers land, partly hundreds of meters to > 1 kilometer. All sites with any type of farmland were locations with private owners that are under the supervision of Embrapa Amazonas. Thus, we could rely on trust of the people and infrastructural support (tractor to pull us out of the dirt, help in hacking trails of several hundred meters length into forest). As mentioned before, all locations were considered highly representative by Embrapa and INPA specialists prior to the first field campaign.

The third point needs to be countered, too. When extreme conditions define seasons of a single year, related effects are generally stronger than under normal conditions. We

were fortunate to encounter such extremes as described in the manuscript. That allows studying climate-related effects and is common practice in many regional climate-change studies.

ASR 2: "there is no background information of upscaling methods. Of major concern is the fact that the authors completely ignored the (often) large uncertainty of upscaled results. In the same manner,as already pointed to the Anonymous Referee #1, the description of statistical methodsand the presentation of results is very incipient."

Our comments 02: The manuscript clearly states that the data were upscaled to one hectare. All necessary information to recalculate that is given in the manuscript. We are more than aware of the risks involved in any type of upscaling. And we claim nowhere that this little exercise goes beyond its scope. My only assumption is that our wording invites related misunderstanding - this can of course be changed. On statistical methods, see our reply to referee 1.

ASR 2: "It is not clear why the authors attempted to compare the soil contents of carbon, nitrogen and sulphur of the Amazon with the soil of other parts of the world. For that reason, the Anonymous Referee #1 stated that "comparing tropical soils with European soils[...] seems illogical". If the authors have any strong argument in favour of doing so,then their argument should be provided in the manuscript. But note that, when makingsuch global comparisons, then the authors should also include other tropical forest in the analysis as well."

Our comments 03: As explained in our replies to referee 1, there is a significant bias in world soil average data (and wide range perception even amongst the science world) that humid tropical soils are XXX. With our "crazy" comparison of the latest empirical CNS data for all Europe, we can show that reality is not quite that blunt. Instead, mineral soils of Europe with radically different climatological and younger geological history show - based on median values - exactly similar values with our from the Amazon basin. Coincidence? We do not think so.

To include other tropical forests as well, as referee 2 suggests, has been attempted. And yet, any search will quickly reveal that very few high-quality data are around. The current attempt of the Max Planck Institute of Biogeochemistry to compile global sub-tropical and tripical data clearly illustrates this. We are in dire need for more and better data for this vast part of the world - and our manscriupt contributes to that.

ASR 2: "Regarding the statistical analysis of the data, I want to stress that the authors should take into account the fact that the sampling sites were chosen in a non-probabilisticmanner and are highly clustered. This means that geostatistical considerations couldbe necessary when analysing the data. Also, some sampling site are very close tomajor cities and/or major rivers, while other sampling sites seem to be very isolated.The authors need to demonstrate that the closeness to major rivers and cities have no(or only minor) influence on the results."

Our comments 04: Exactly that approach has been taken in our statistical analysis - see reply to referee 1. Solveig Pospiech and Gerald van den Boogart took great care to test exactly that, too. Just generally speaking: Proximity to major rivers or to larger urban centres of our locations is just as balanced as it is in reality - see figure 1. As soon as a locations is far enough not to get inundanted in the rainy season (=varzea or igapo) the absolute distance from a river is irrelevant. It is more important, which lithology underlies the locations and what land cover there is. With urban structures, it is a similar story: As soon as there is no more direct urban impact (see e.g. deep within Reservatorio Adolfo Ducke, near Manaus), there is no noticable urban impact in soil chemistry.

ASR 2: "Finally, given that the present study was funded by government agencies, both Brazilian and German, the authors should present a long term data management plan. Forinstance, this could be done though its publication in a research data repository."

Our comment 05: There is, but is this ever part of an individual manuscript? We provide all manuscript-related data to the readers as electronic supplement. And all data are

and will be available to anyone interested with all meta-data.

Final words of ours: We understand that some of our wording will have provoked mis-understandings and that we should be more explicit on some details in methodology.

Sincerely and for all co-authors

Jörg Matschullat
* * *

---

## Author Comment (AC3) · 30 Jul 2019

Dear Hannes Reuter, dear Editor,

I was made aware that it might be helpful to further substantiate our comment regarding the bias of CNS and C-isotopic data in Europe and North America versus subtropical and tropical regions of the world (which dominate the continental surface):

https://international-soil-radiocarbon-database.github.io/ISRaD/

While this site focuses on soil radiocarbon, it is quite representative for the general knowledge gaps in soil carbon, nitrogen (and sulfur). One of the related problems

is that standard elemental analyzers do not deliver nitrogen values, and many face challenges with lower limits of determination in sulfur quantification.

Our lab is the only one in Europe that successfully passed the quality-demands of EuroGeoSurveys for the GEMAS atlas project and more recently now for the Austrlian Geoscience mapping programme NGSA (not yet published).

Sincerely

Jörg Matschullat